# Dissipationless transport signature of topological nodal lines

Arthur Veyrat[1,2,3] ✉, Klaus Koepernik [1,2], Louis Veyrat [1,2,4], Grigory Shipunov [1,2], Iryna Kovalchuk [1,5], Saicharan Aswartham [1,2], Jiang Qu [1,2], Ankit Kumar [1,2], Michele Ceccardi[6,7], Federico Caglieris [7], Nicolás Pérez[1], Romain Giraud [1,2,8], Bernd Büchner [1,2,9], Jeroen van den Brink [1,2,9], Carmine Ortix [10] ✉ & Joseph Dufouleur [1,2,11] ✉

Topological materials, such as topological insulators or semimetals, usually not only reveal the non-trivial properties of their electronic wavefunctions through the appearance of stable boundary modes, but also through very specific electromagnetic responses. The anisotropic longitudinal magnetoresistance of Weyl semimetals, for instance, carries the signature of the chiral anomaly of Weyl fermions. However for topological nodal line semimetals—materials where the valence and conduction bands cross each other on one-dimensional curves in the three-dimensional Brillouin zone—such a characteristic has been lacking. Here we report the discovery of a peculiar charge transport effect generated by topological nodal lines in trigonal crystals: a dissipationless transverse signal in the presence of coplanar electric and magnetic fields, which we attribute to a Zeeman-induced conversion of topological nodal lines into Weyl nodes under infinitesimally small magnetic fields. We evidence this dissipationless topological response in trigonal $PtBi_2$ persisting up to room temperature, consistent with the presence of extensive topological nodal lines in the band structure of this non-magnetic material. These findings provide a pathway to engineer Weyl nodes by arbitrary small magnetic fields and reveal that bulk topological nodal lines can exhibit non-dissipative transport properties.

The electronic band structure of a bulk material can feature isolated degeneracy points where electronic states with different spin, orbital or sublattice quantum numbers possess the same energy and crystalline momentum **k**. In materials lacking either inversion or time-reversal symmetry, such degeneracies can be simply twofold while occurring at generic points in the three-dimensional Brillouin zone. The electronic bands in the vicinity of the nodes then generally resemble the energetic dispersion of massless relativistic particles governed by the Weyl equation[1]. Weyl nodes represent monopoles of the Berry flux and are thus characterized by a well-defined topological charge. This non-trivial bulk topology is manifested in a very specific spectroscopic signature: the presence of surface Fermi arcs connecting Weyl points

[1]Leibniz Institute for Solid State and Materials Research (IFW Dresden), Helmholtzstraße 20, Dresden, Germany. [2]Würzburg-Dresden Cluster of Excellence ct.qmat, Dresden, Germany. [3]Laboratoire de Physique des Solides (LPS Orsay), 510 Rue André Rivière, Orsay, France. [4]CNRS, Laboratoire National des Champs Magnétiques Intenses, Université Grenoble-Alpes, Université Toulouse 3, INSA-Toulouse, EMFL, Toulouse, France. [5]Kyiv Academic University, Kyiv, Ukraine. [6]Department of Physics, University of Genoa, Genoa, Italy. [7]CNR-SPIN Institute, Genoa, Italy. [8]Université Grenoble Alpes, CNRS, CEA, Grenoble-INP, Spintec, Grenoble, France. [9]Department of Physics, TU Dresden, Dresden, Germany. [10]Dipartimento di Fisica "E. R. Caianiello", Universitá di Salerno, Fisciano (SA), Italy. [11]Center for Transport and Devices, TU Dresden, Dresden, Germany. ✉e-mail: arthur.veyrat@universite-paris-saclay.fr; cortix@unisa.it; j.dufouleur@ifw-dresden.de

with opposite chirality[2,3]. The characteristic electromagnetic response of Weyl quasiparticles is instead connected to their chiral anomaly[4,5]. This causes a strong in-plane anisotropic magnetoconductivity that can be directly probed through measurements of the planar Hall effect (PHE)[6–8]: the appearance, in the presence of coplanar electric and magnetic fields, of a transverse voltage with $\pi$-periodic angular dependence. Weyl quasiparticles may also be evidenced in transport experiments through other effects, such as the unconventional Hall effect[9].

Point-group symmetries can also stabilize twofold degenerate closed lines in the three-dimensional Brillouin zone[10]. When appearing at mirror-invariant planes, such nodal lines are characterized by a bulk $\mathbb{Z}_2$ topological invariant[11,12]. Although they are often accompanied by "drumhead" surface states[13–15], topological nodal lines (TNLs) lack a genuine bulk-boundary correspondence: the relevant surfaces naturally break the protecting mirror symmetry[16]. Additionally, some electromagnetic responses characteristic of TNL-semimetals have been identified, but only in specific cases, such as, for instance, in the quantum limit[17], making the physical consequences of the bulk topology completely hidden. Here, we unveil a peculiar charge transport effect associated with mirror symmetry-protected TNLs in trigonal crystals: an anomalous planar Hall effect (APHE) that is odd in magnetic field, does not contribute to the dissipated power[18,19], and is measurable in the linear transport regime. We identify trigonal-PtBi$_2$ as an ideal material platform because of the presence of a large number of TNLs on its three vertical mirror planes, which makes the anomalous planar Hall effect particularly robust and survives up to room temperature.

## Results and discussion

In low-dimensional systems, such as LaAlO$_3$/SrTiO$_3$ oxide interfaces, the occurrence of an APHE has been reported and is due to a Zeeman-induced modification of local concentrations of the out-of-plane Berry curvature, which, when integrated over momenta, becomes non-vanishing[20]. An APHE has also been reported in VS$_2$-VS heterostructures[21]. The mechanism we find to be at work in TNLs is completely different in nature. It is caused by a Zeeman-induced conversion of TNLs into Weyl nodes that generalizes the fusion of Weyl nodes into nodal lines predicted to occur in mirror-symmetric systems[12]. The important point is that magnetic fields that break the mirror symmetry protecting the TNLs lead to a non-local conversion of the TNL into Weyl nodes of opposite chirality, meaning that they are separated in momentum space by a vector that has components parallel to the mirror plane. The extraordinary feature of this $k$-space separation in momentum space is that it survives even for infinitesimally small magnetic fields and can be as large as the diameter of the TNL. This conversion and its properties can be qualitatively captured using a simple two-band low-energy model (Supplementary Note M). This generally induces large momentum regions of non-zero Chern number, thus generating an anomalous planar Hall effect (APHE) already at infinitesimal magnetic field, with a much larger amplitude than that which would be obtained solely from the Zeeman-induced displacement of Weyl nodes (Supplementary Notes J and M). Consider, for simplicity, a single pair of TNLs related to each other by time-reversal symmetry and protected by a vertical mirror plane, which, without loss of generality, we set as $\mathcal{M}_x$ (see Fig. 1a). With an infinitesimally small magnetic field along the $\hat{y}$ direction, the TNLs convert into two field-induced pairs of Weyl nodes, each of which has a separation in $k_z$ comparable to the TNL dimension, generating a dissipationless (i.e., without diagonal components) antisymmetric Hall conductance $\sigma_{yx}$ (i.e., an APHE). The system can be viewed in fact as a collection of two-dimensional $\{k_x, k_y\}$ insulating layers[22] parameterized by $k_z$ and characterized by a local Chern number $c(k_z)$ (see Fig. 1a) that is changed by the topological charge of each Weyl node. The non-local Zeeman-induced conversion of the TNLs into Weyl

nodes then leads to a net $\sigma_{yx} = \int c(k_z)dk_z$ and to Hall voltages that lie in the same plane as the applied current and the external magnetic field, precisely in a configuration where the conventional Hall effect is absent. Such APHE thus represents a characteristic electromagnetic response of TNLs.

Additionally, Onsager's relations[23] enforce the transversal APHE conductance to be odd under a magnetic field reversal and thus compatible only with an out-of-plane threefold rotational symmetry (see Fig. 1b). This property, along with its non-dissipative character, makes the APHE experimentally distinguishable from a planar Hall effect. First, a PHE, such as that associated with the Berry curvature of the Weyl nodes, is characterized by a $\pi$-periodic oscillation of both the longitudinal $R_{xx}$ and transverse resistance $R_{yx}$, when the magnetic field is rotated in-plane while keeping the current direction fixed, with a $\pi/4$ offset between them (see Fig. 1b and "Methods")[6,7]. The longitudinal resistance $R_{xx}$ is, moreover, maximized when the magnetic field and current are aligned, while the transverse resistance $R_{yx}$ vanishes in this configuration. As a result, a twofold symmetric PHE aligned with current direction can easily be disentangled from a threefold-symmetric APHE aligned with crystalline axes (see Fig. 1b). We note that the APHE generally implies finite transversal conductance even for a magnetic field parallel to the electric field, a seldom-seen situation that has already been reported in the case of non-magnetic[24] and magnetic materials[25], the latter case exhibiting a threefold symmetry. More importantly, the APHE is non-dissipative, i.e it is not associated with any corresponding longitudinal signal. This allows it to be distinguished unambiguously from any potential threefold symmetric PHE due to magneto-crystalline anisotropies, which would be associated with a corresponding AMR.

We now show that both these effects can be probed in the layered van der Waals material PtBi$_2$, which has recently been characterized as a non-magnetic type I Weyl semimetal. PtBi$_2$ also exhibits sub-Kelvin 2D-superconductivity and a BKT transition in nanostructures[26], as well as higher-temperature surface superconductivity[27] with superconducting topological Fermi arcs[3]. The crystallographic point-group symmetry of PtBi$_2$ is $\mathcal{C}_{3v}$ that is comprised of a threefold axis and three vertical mirror planes $\mathcal{M}$ (Fig. 1c)[28], and is compatible with the appearance of an APHE. When an in-plane magnetic field is perpendicular to a mirror plane, the anomalous Hall conductivity (AHC) $\sigma_{yx}$ must vanish. Conversely, when the field is parallel to a mirror plane, $\sigma_{yx}$ is maximal[18]. This results in a $2\pi/3$ periodic angular dependence (see Fig. 1c) that can be detected in practice with magnetotransport measurements.

We focus our study on a 70 nm thick nanostructure (see magnetotransport measurement schematic in Fig. 1d) investigated up to 14T, and temperatures from 5 K up to 300 K (two additional structures showed similar behavior, see Supplementary Notes F and G). The results are shown in Fig. 2. First, in exfoliated nanostructures of PtBi$_2$, we systematically observed a PHE. At $T = 100$ K and $B = 14$ T, a pronounced $\pi$-periodic oscillation is clearly visible in both $R_{xx}$ and $R_{yx}$ (Fig. 2a), with the expected $\pi/4$ angular shift between them (Fig. 2b). The PHE is already visible at magnetic fields as low as 1 T (see Supplementary Fig. 2). The angular positions of the maxima of $R_{xx}$ are consistent with the expected current orientation in the sample (see "Methods"). The PHE is very robust with temperature, and for $B = 14$ T it can be evidenced up to room temperature (Fig. 2c). The presence of a strong PHE in the non-magnetic PtBi$_2$ reveals the large BC present in the material, giving significant experimental indications of its Weyl nature, and confirming predictions from previous band structure calculations[26].

The main experimental result of this work is the evidence of an APHE in our measurements, which appears as a small deviation from the PHE in Fig. 2a–c. In order to evidence the APHE in our data, we remove a $\pi$- and $2\pi$-periodic background from each measurement (in red in Fig. 2a, b, and Supplementary Note B). Removing each signal separately would be similar to antisymmetrizing (resp. symmetrizing)

the data in the magnetic field, albeit with more control over exactly which terms are removed. The resulting residues are depicted in Fig. 2d–f. At 14T, $2\pi/3$-periodic oscillations, which, contrary to the standard PHE signal, are antisymmetric in B, are clearly visible in the transverse resistance residues $\Delta R_{yx}$ at 100 K (Fig. 2d, e) and can be fitted with cosine fits (Fig. 2e, in green). This $2\pi/3$-periodic signal appears above $B = 4$ T, at a constant angular position, and its amplitude increases with magnetic field (Supplementary Note D). Importantly, no associated $2\pi/3$-periodic signal is visible in the longitudinal resistance residues $\Delta R_{xx}$, up to the highest fields and down to the lowest temperatures (Supplementary Fig. 4). This lack of longitudinal component is critical in distinguishing the dissipationless APHE from any conventional, dissipative PHE, which would necessarily be associated with an AMR. Remarkably, the APHE is very robust in temperature, as the oscillations remain visible from 5K up to room temperature, as shown in Fig. 2f.

The field and temperature dependence of the APHE signal obtained from the cosine fits are presented in Fig. 2g, h. At 5K, the APHE signal is visible above 4T, and increases linearly with field. This fit yields a critical magnetic field $B_c \sim 2.8$ T for the appearance of an APHE.

At 14T, the APHE signal remains constant in temperature below 20 K, similar to the PHE and the longitudinal resistance (see Supplementary Note C). Above this temperature, it decreases following an exponential decay law $A^{APHE}(T) - A^{APHE}(T=0) \propto e^{-k_B(T-T_c)/\Delta}$ with $T_c \sim 30$ K and an energy scale $\Delta \sim 6$ meV. The existence of an onset field for the APHE could be explained by the fact that, while the TNLs are already topologically gapped at infinitesimal fields, this gap vanishes at low magnetic fields, and the APHE would only become visible when it exceeds the thermal energy broadening somewhere along the TNL. Similarly, the exponential decay of the signal in temperature could come from thermal broadening through the TNL gap opened by the magnetic field (Supplementary Note N). This is supported by the temperature shift on the onset field evidenced in sample D3 (see Supplementary Fig. 8). In this case, the energy scale of the decay, $\Delta$, would be linked to the gap of the TNLs.

We next demonstrate that the APHE experimentally observed is consistent with the presence of TNLs in PtBi$_2$ by performing full-relativistic electronic band structure calculations, with and without a magnetic field. In the absence of Zeeman coupling, the material features 3 groups of Weyl nodes, which, due to the concomitant presence

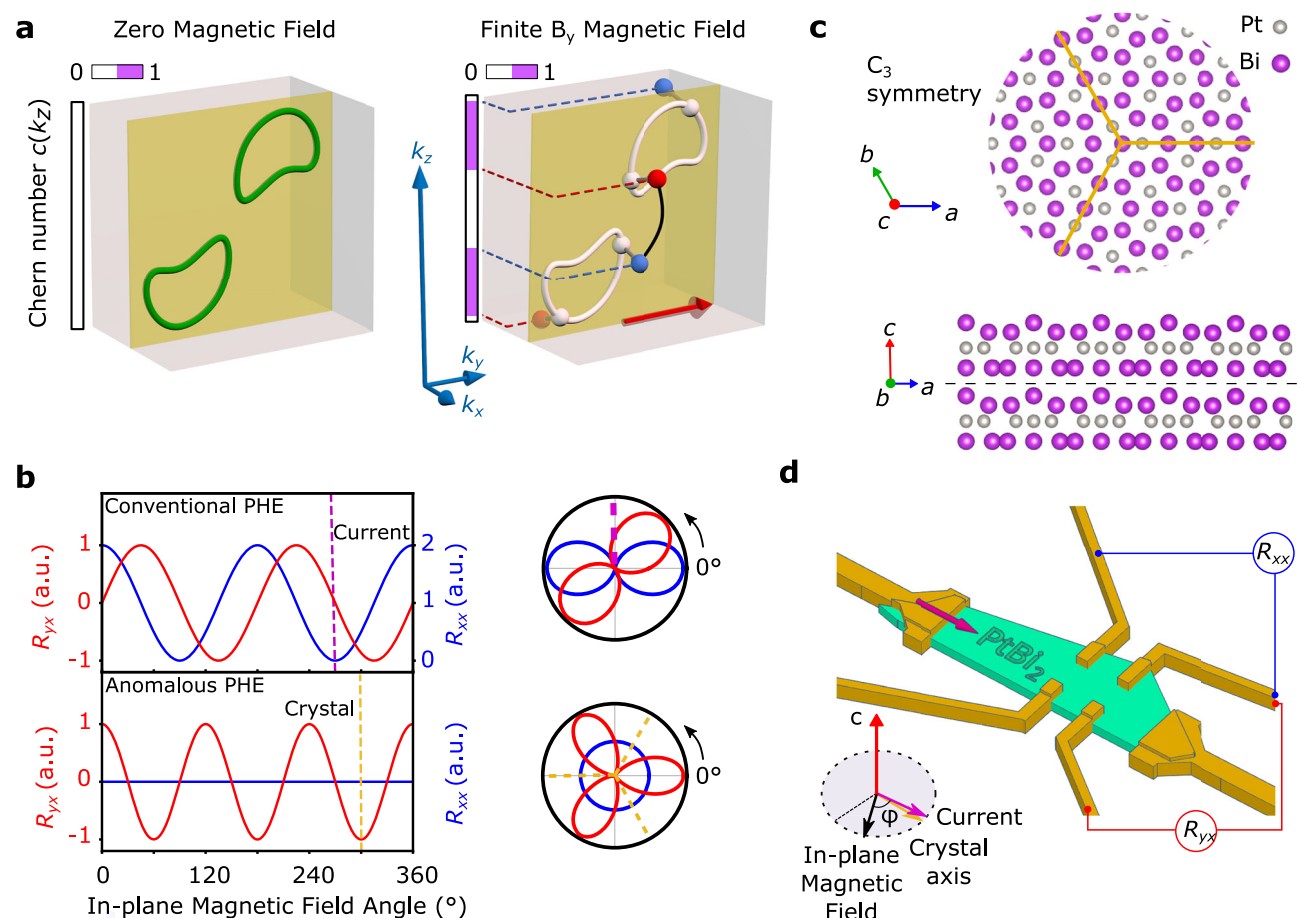

**Fig. 1 | Anomalous planar Hall effects in PtBi$_2$. a** Generation of an anomalous Hall conductance in nodal line semimetal systems. In a nodal line semimetal, the nodal lines do not contribute to the Chern number at zero magnetic field (left panel). Under a finite external magnetic field (right panel, red arrow), the nodal lines split into pairs of Weyl nodes of opposite chiralities (red and blue). These pairs can appear anywhere on the nodal lines (white line), including with a significant $k_z$ separation. This leads to potential large $k_z$ ranges of non-zero $c(k_z)$ (pink color in the rectangle), inducing a large AHC at finite field. **b** Typical angular dependence of the conventional (top panel) and anomalous (bottom panel) planar Hall effects, in Cartesian (left) and polar (right) coordinates. For the conventional PHE, both the longitudinal (anisotropic magnetoresistance, $R_{xx}$, blue) and transverse (planar Hall effect, $R_{yx}$, red) resistances exhibit a $\pi$-periodic angular dependence, with a $\pi/4$-offset between them. The origin of the oscillation is set by the direction of the electric field (current). For the APHE, the angular dependence is $2\pi/3$-periodic, with origin set by the crystal directions, and is not associated with any AMR. **c** Crystal structure of trigonal-PtBi$_2$, with layered nature and in-plane $\mathcal{C}_3$-symmetry highlighted. **d** Sample configuration. The pink arrows indicate the direction of the current. The yellow arrow corresponds to a specific crystal orientation and the black arrow indicates the direction of the magnetic field. The angle $\varphi$ refers to the orientation of the in-plane magnetic field **B**.

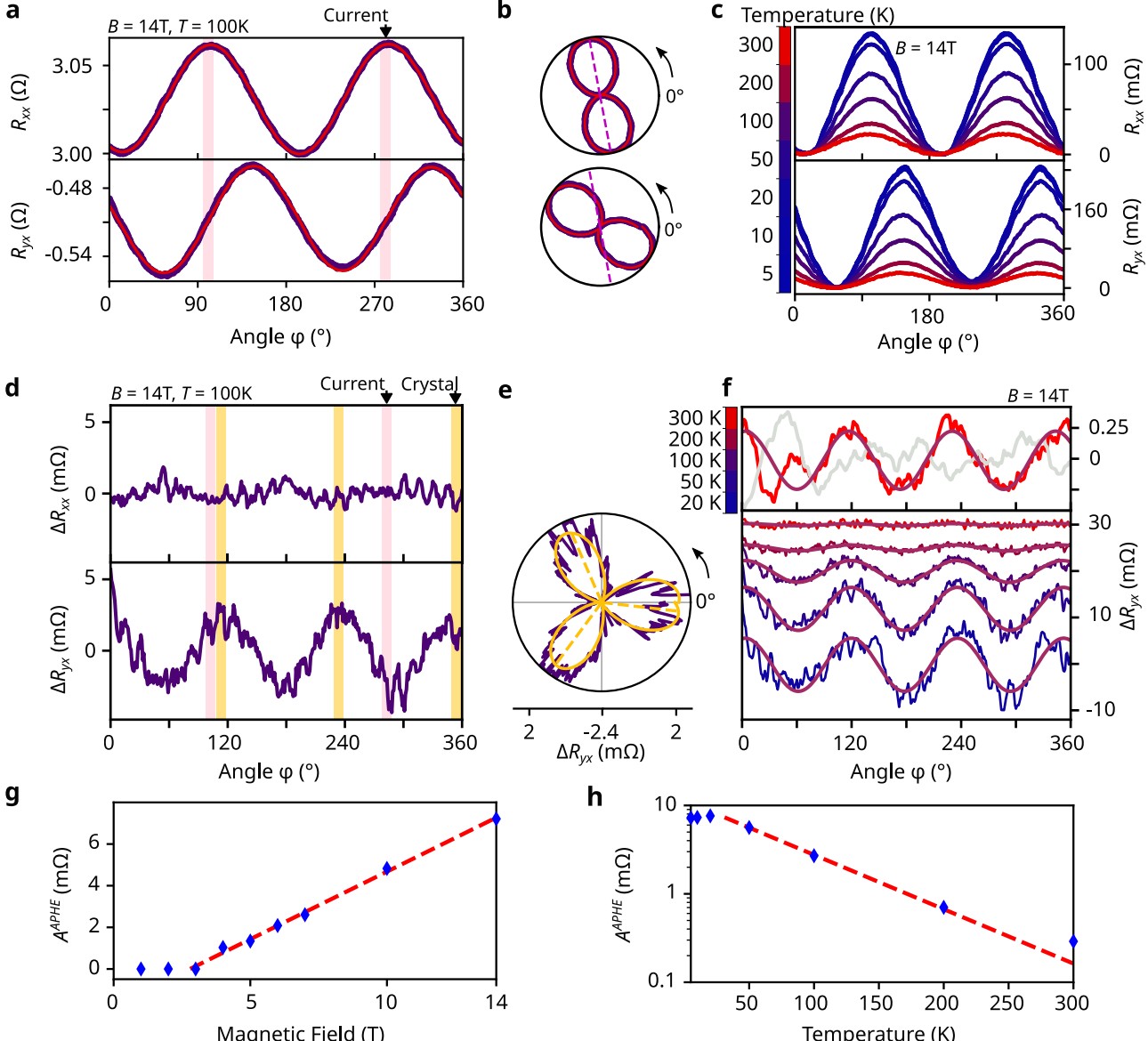

**Fig. 2 | Standard and Anomalous Planar Hall effect in PtBi₂. a, b** Angular dependence of $R_{xx}$ and $R_{yx}$ at 14T, 100K in Cartesian (**a**) and polar coordinates (**b**). The fits with Equation 1 (Methods) are shown in red in **a** and **b**. The radial axis has the same range as in **a**. The pink bars in **a** and the pink dashed line in **b** show the current direction estimated from the fits, with a ±5° width. **c** Angular dependence of $R_{xx}$ and $R_{yx}$ at different temperatures from 5 to 300 K, at 14T. The curves are vertically shifted for clarity. **d, e** Angular dependence of the residues $\Delta R_{xx}$ and $\Delta R_{yx}$ from the data in **a** after a background removal (see Supplementary Note B), in cartesian (**d**) and polar coordinates (**e**). A $2\pi/3$-periodic signal is clearly visible in $\Delta R_{yx}$. The pink and green bars show the previously estimated current direction and the crystal direction estimated from the fits to Supplementary Materials eq. S5, respectively, with a ± 5° width. In **e**, the fit to Supplementary Materials eq. S5 is shown in green. **f** Bottom: angular dependence of $\Delta R_{yx}$ at 14T for $T$ = 20, 50, 100, 200, and 300 K, with fits to Supplementary Materials eq. S5 shown in green. The curves are vertically shifted for clarity. Top: Angular dependence of $\Delta R_{yx}$ at 14 T, 300 K, with fit in green. The 300K data was smoothed over 31° for visibility. The corresponding $\Delta R_{xx}$ signal is plotted in a gray line with the same scale for comparison, and shows no visible periodic signal. **g** Field dependence of the APHE amplitude $A^{APHE}$, showing a linear dependence above $B_c$ ~ 2.8 T. The dashed line indicates the best linear fit for $B$ > 2.5 T. **h** Temperature dependence of $A^{APHE}$, showing an exponential decay above $T_c$ ~ 30 K, with an energy scale $\Delta$ ~6 meV. The dashed line corresponds to the best exponential fit for $B \gtrsim$ 30 T.

of time-reversal symmetry and the threefold rotation symmetry, all come with multiplicity twelve. These groups of Weyl nodes appear above the Fermi energy, with the one lowest in energy being at around 45.3 meV above $E_F$, in agreement with a previous study[26]. In each of these groups, pairs of Weyl nodes of opposite chirality are connected by a vector perpendicular to the mirror plane. The presence of in-plane magnetic field leads to a movement of the Weyl nodes in all momentum directions due to the low residual symmetry: the system possesses at most a vertical $\mathcal{M}' = \mathcal{M} \times \Theta$ symmetry (with $\Theta$ time-reversal) when the magnetic field is parallel to the mirror plane $\mathcal{M}$. However, the Weyl

node displacement is proportional to the strength of the applied magnetic field and remains relatively small at laboratory-accessible fields. The resulting anomalous planar Hall conductance from this displacement is therefore expected to be vanishingly small (Supplementary Note J).

The situation is completely different for the three pairs of nodal loops revealed by our calculations, which lie in the vertical mirror planes of PtBi₂ and are reminiscent of nodal chain semimetals[12]. An infinitesimal magnetic field leads to the conversion of the TNLs, each into 6 Weyl nodes [see Fig. 3a]. Since in PtBi₂ the degeneracy loops do

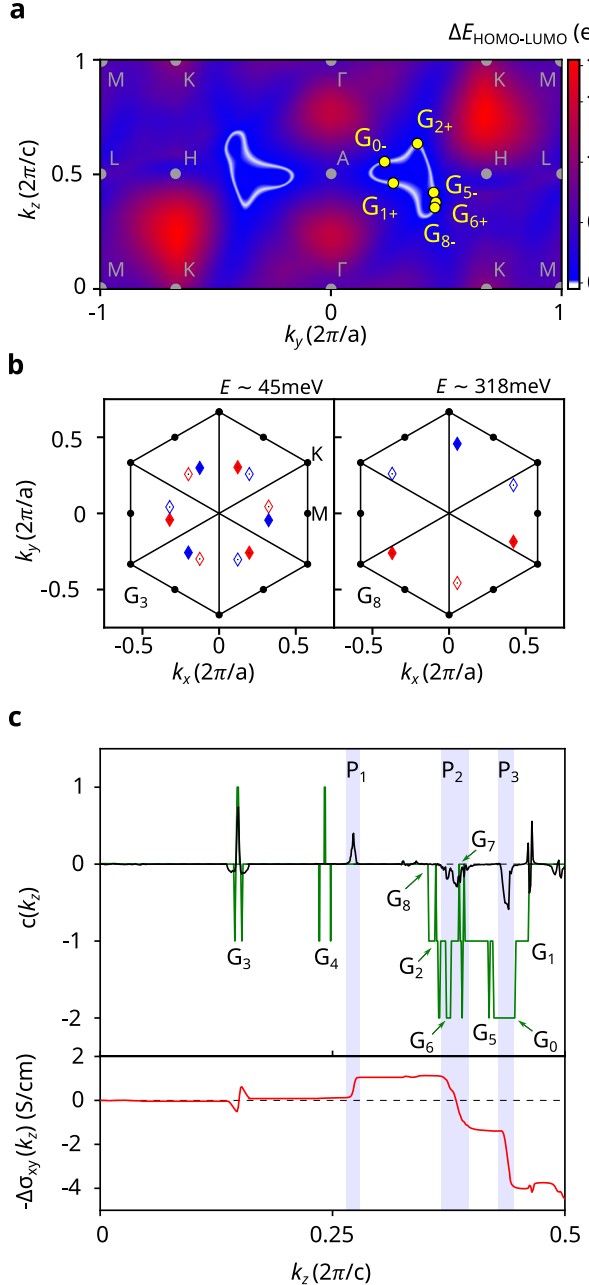

**Fig. 3 | Nodal lines and the origin of the anomalous Planar Hall effect in PtBi$_2$.**
**a** Energy gap $\Delta E$ between HOMO and LUMO bands in the $k_y$, $k_z$ (mirror) plane. The nodal loops ($\Delta E = 0$) appear in white. When $B \neq 0$, each nodal loop splits into 6 Weyl nodes (WN, yellow points), forming 6 groups of 6-WN. The signs denote the chiralities. **b** Two groups of WN of HOMO-LUMO for a Zeeman energy $E_Z = 14$ meV: $G_3$ is the 12-fold set of WNs closest to $E_F$ already present at $B = 0$, and $G_8$ is one of the six 6-fold groups mentioned above. The average energies of the groups are shown. Red (blue) markers denote positive (negative) chirality, while full (empty) markers denote the positive (negative) $k_z$ position of the WN ($G_3$: $k_z$ -$\pm$ 0.149, $G_8$: $k_z$ -$\pm$ 0.358). Solid lines represent the mirror planes, while the dots show the high-symmetry points. **c** (Top) Chern number $c(k_z)$ in an ideal (green, full HOMO) and a more realistic (black, $E_F = E_{G_3} = 45.3$ meV) case, with a a Zeeman energy $E_Z = 14$ meV. In the ideal case, the Chern number jumps discretely by $\pm 1$ at each WN, while the variation is smoothed out in the realistic case. (Bottom) Anomalous Hall conductivity$-$ $\Delta\sigma_{xy}(k_z)$ calculated from the Chern signal in the realistic case (in black above). The 12 WNs from $G_3$ at low $k_z$ contribute very little to the AHC, as the Berry curvature they generate is nearly compensated. Most of the AHC comes from 2 peaks in the Chern number at higher $k_z$, $P_2$, and $P_3$ (shown in blue). A third peak at lower $k_z$, $P_1$, attenuates the total AHC amplitude, and is found to correspond to WNs from nodal lines below the HOMO band (see Supplementary materials sec. K). Only the $k_z > 0$ dependences are shown, as $c(k_z)$ is even and $\Delta\sigma_{xy}$ is odd in $k_z$.

an applied magnetic field parallel to one of the vertical mirror planes of the material (see Fig. 3c, in green). We note that $E_Z$ was chosen at such a value for numerical resolution purposes, and does not correspond to our experimental conditions. As expected, the twelvefold groups of precursive Weyl nodes do not give a large contribution to the anomalous planar Hall conductance.

We have also performed a realistic calculations assuming that bands are filled with a Fermi level set at the energy of the Weyl nodes group at 45.3 meV above the non-magnetic $E_F$ (black line in Fig. 3c). We find that the local Chern signal (see Supplemental Material) of the twelvefold groups is washed out whereas the jumps due to the sixfold groups originating from the conversion of the TNLs are smoothed. However, the realistic contribution to the anomalous planar Hall conductance (red line in Fig. 3c) is not only due to these TNLs but also derives from the existence of additional nodal lines (Supplementary Note K) that we find in lower valence bands and also undergo conversion into Weyl nodes in the presence of planar magnetic fields. Quantitatively, the contribution of the twelvefold groups to the APHE (see Supplementary Note J) is estimated to be more than two orders of magnitude smaller than the contribution of the TNL-induced WNs. Moreover, we note that another possible origin of the APHE, the orbital intrinsic planar Hall effect, is forbidden in systems with $\mathcal{C}_{3v}$ point-group symmetry[29]. This further demonstrates that the anomalous planar Hall conductance of PtBi$_2$ is a direct electromagnetic response of TNLs.

To sum up, we measured in 3D nanostructures of the non-magnetic 3D Weyl semimetal PtBi$_2$, beyond a conventional PHE, a very robust APHE with a signature $2\pi/3$-periodic oscillation in $\Delta R_{yx}$ and an absent dissipative signal in $\Delta R_{xx}$. This APHE is consistent with the presence of topological nodal lines in PtBi$_2$'s band structure, through a non-local conversion to Weyl nodes under a magnetic field. This mechanism can be generally used to engineer Weyl nodes in materials featuring mirror symmetry-protected nodal lines by means of arbitrarily small magnetic fields. Our observations also establish the anomalous planar Hall effect as an efficient magnetotransport tool to reveal the presence of TNLs in trigonal semimetals, which could so far only be characterized through spectroscopy measurements. Demonstrating the presence of topological features through transport is especially interesting in PtBi$_2$, where 2D superconductivity was recently reported[26], and a recent ARPES study further found the superconducting weight to be localized on the topological Fermi arcs[3], opening perspectives for possible topological superconductivity.

not occur at fixed energies, these Zeeman-induced Weyl nodes form different groups that are separated in energy$-$we remark that one of these groups has a complex evolution as the magnetic field is increased as it directly combines with one of the preexisting twelvefold Weyl node groups at $B = 0$ (see Supplementary Note I). For a magnetic field parallel to one mirror plane, each of these groups is sixfold, with pairs of opposite chirality related by the combined $\mathcal{M}'$ symmetry. This immediately implies that an isolated group yields a sizable contribution to the anomalous planar Hall conductance. The latter can be computed in an "ideal" case by simply assuming that the Weyl nodes are all at the Fermi level, in which case each Weyl node provides a unit change to the Chern number of the insulating $(k_x, k_y)$ layers. The distribution of the Weyl nodes in two trios (see Fig. 3b) at nearly opposite values of $k_z$ demonstrates that a large contribution to the anomalous planar Hall conductance can be expected from the sixfold groups of Zeeman-induced Weyl nodes. This is verified by a direct calculation of the local Chern number of the full system assuming, as before, all Weyl nodes are at the Fermi energy and a Zeeman energy $E_Z = 14$ meV from

## Methods

### Sample preparation

High-quality single crystals of $PtBi_2$ were grown using the self-flux method[28]. These crystals were mechanically exfoliated to obtain thin flakes, with widths exceeding 10 μm and thicknesses ranging from a few dozen to a few hundred nanometers. The flakes were contacted with Cr/Au using standard e-beam lithography techniques. Prior to the metal deposition, a small Ar-etch was performed to eliminate any surface oxidation. The main sample used in this study is denoted as $D1$ (70 nm thick), and supplementary information includes corroborating results for a second sample, D2 (126 nm thick) and a third sample, D3 (320 nm thick). In a previous study[26], the two-dimensional superconductivity of these samples was studied in details at sub-Kelvin temperatures. Here, we focus on measurements performed above 1K, above the superconducting transition. No evidence of significant aging effects was observed between the two studies, as indicated by the similar residual resistance ratio $RRR = R(300K)/R(4K)$ (Supplementary Note H).

### Measurement setup

The measurement configuration consists of a standard Hall-bar geometry. A current is injected between the source and the drain as depicted in Fig. 1d. Longitudinal and transverse resistances (indicated in red and black, respectively) are measured along and across the sample relative to the current orientation.

**Measurement set-up for samples D1 and D2.** Measurements were conducted in a Dynacool 14T PPMS using an insert equipped with a mechanical 2D rotator. By rotating the sample with the rotator, the angle $\varphi$ between the fixed-axis magnetic field and the applied current can be adjusted over a full range of 360° (with $\varphi$ the angle between the magnetic field and the electric field. The resistances were measured using external lock-in amplifiers, with an AC current of 100 μA at a frequency of 927.7 Hz, with an integration time of 300 ms. At such low currents, no thermal effects are expected. For sample $D1$, for measurements taken at $T = 5$ K and $B = 1, 2, 3, 4, 5, 6, 7, 10$ T, as well as at $B = 14$ T and $T = 5, 10, 20, 50, 300$ K, 10 points were measured at each angular position, taking the averaged value of the resistance. The angular step for each measurement was 1°. All measurements on sample $D2$ were conducted with the same parameters. For sample $D1$, more precise measurements were taken at $T = 5$ K and $B = 14$ T, as well as at $B = 14$ T and $T = 100, 200$ K, with an averaging over 40 measurement points at each angular position. The angular step for each measurement was 0.5°, and the results were interpolated with a step of 1°, to perform the analysis in the same way for each pair of (B,T) parameters.

At low temperature ($T \leq 20$K) the first oscillation of the APHE ($0° \leq \varphi \leq 120°$) is not fully visible even at 14T, although it is consistently observed for $T \geq 50$ K (see at $T = 100$ K in Fig. 2a and Supplementary Notes E and D). This partial suppression of the signal likely stems from the mechanical rotator: When the stepper-motor at the top of the measurement stick turns by a small angle (in our measurements, the angular step is 1°), the mechanical rotator in the cryostat will move by an inconsistent angle (around the target step, e.g. 1°). As we measure the angle of the rotator at the top of the stick, and not the actual angle of the sample at the bottom, this creates small deviations of the PHE signal away from a $\pi$-periodic oscillation. When a $\pi$-periodic background is removed from the data (to evidence the APHE signal), these deviations are carried to the residues, and can corrupt the signal. These artifacts are reproducible and decrease with temperature, as would be expected with mechanical rotator inconsistencies.

**Measurement set-up for sample D3.** Measurements were conducted in a VTI equipped with a 3D-piezorotator and a large bore 14T magnet. In this work, we show in-plane rotation measurements. By rotating the sample with the rotator, the angle $\varphi$ between the fixed-axis magnetic field and the applied current can be adjusted over a 180° range (with $\varphi$

the angle between the magnetic field and the electric field). In order to have the full 360° range, we flip the orientation of the sample by 180° along the perpendicular axis. This is equivalent to reverse the direction of the magnetic field. The resistances were measured using external lock-in amplifiers, with an AC current of 500 μA at a frequency of 331 Hz, with an integration time of 300 ms. For sample $D3$, for measurements taken at $T = 3, 20, 100$ K for fields ranging between 1 T and 14 T. Contrary to the measurement of D1 and D2, a single point was measured at each angular position.

### Planar Hall effect

The contributions of the planar Hall effect/anomalous magnetoresistance to the longitudinal resistivity $\rho_{xx}$ and transverse resistivity $\rho_{yx}$ obey the following angular dependence[6,7]:

$$\begin{aligned} \rho_{xx}^{AMR}(\varphi) &= \rho_\perp - \Delta\rho\cos^2\varphi, \\ \rho_{yx}^{PHE}(\varphi) &= -\Delta\rho\cos\varphi\sin\varphi, \end{aligned} \quad (1)$$

with $\Delta\rho = \rho_\parallel - \rho_\perp$ the amplitude of both the PHE and the AMR; $\rho_\parallel$ and $\rho_\perp$ the resistivities when **B** is respectively along and perpendicular to the electrical field (current); and $\varphi$ the angle between the magnetic and electric fields (i.e., current lines) in the sample. The PHE signal is therefore characterized by $\pi$-periodic oscillations for both $\rho_{xx}$ and $\rho_{yx}$ (when rotating the magnetic field in the sample's plane, with a fixed current) with the same amplitude, with a $\pi/4$ offset between the two. The maxima of $\rho_{xx}$ correspond experimentally to the orientation of the current in the sample (Supplementary Note A).

### Computation details

We performed a full-relativistic non-magnetic calculation using the full potential local orbital (FPLO) code[30] version 22.01 within the generalized gradient approximation (GGA)[31]. The lattice parameters can be found in Supplementary Note L. From the DFT result a 72-band Wannier function (WF) model was extracted consisting of Bi$6p$ and Pt$6s5d$ type WFs. A constant magnetic field Zeeman term $H^{Zeeman} = B\mu_B\langle S\rangle$ was added to the model using the WF representation of the spin operators $\langle S\rangle$.

The anomalous planar Hall signal $\Delta\sigma_{yx}$ can be computed by calculating the Chern signal $c(k_z) = \frac{1}{2\pi}\int F_z(k)dS$ for a number of $k_z$-planes with subsequent integration over $k_z$. For an assumed constant homo $c(k_z)$ can be obtained by a plaquette type integration (see Supplementary Note J and ref. 32) and with a Fermi level by a simple Riemann-sum integral.

## Data availability

The data that support the findings of this study are available from the corresponding authors.

## Code availability

The code that was used to calculate the band structure is available from the corresponding authors upon request.

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

## Acknowledgements

A.V. acknowledges funding from the European Research Council (ERC) under the European Union's Horizon 2020 research and innovation program (grant Ballistop agreement no. 833350). S.A. acknowledges the financial support of (DFG) through the grant AS 523/4-1. C.O. acknowledges support from the MAECI project "ULTRAQMAT". L.V. was supported by the Leibniz Association through the Leibniz Competition. This work was supported by the Deutsche Forschungsgemeinschaft (DFG, German Research Foundation) through the Sonderforschungsbereich SFB 1143 and under Germany's Excellence Strategy through the Würzburg-Dresden Cluster of Excellence on Complexity and Topology in Quantum Matter—ct.qmat (EXC 2147, project-ids 390858490 and 242021).

## Author contributions

A.V. and J.D. conceived the project. A.V. designed and fabricated the samples D1 and D2, conducted the measurements and analyzed the data in these samples with input from L.V., N.P. and J.D. The theory was developed by C.O., K.K. and J.v.d.B. Computation of the band structure was done by K.K. J.Q., A.K., M.C. and J.D. fabricated the sample D3, conducted the measurements and analyzed the data in this sample. The crystals were grown by G.S., I.K. and S.A. F.C., R.G. and B.B. contributed to the discussion of the data. All authors participated in the interpretation of the results and writing of the manuscript. C.O. and J.D. supervised the project.

## Funding

## Competing interests

The authors declare no competing interests.
