## [Transparent Peer Review file · Nature Communications]

Dissipationless transport signature of topological nodal lines

Corresponding Author: Dr Joseph Dufouleur

Version 0:

Reviewer comments:

Reviewer #1

(Remarks to the Author)

I have read the revised manuscript and the response of the authors submitted to Nature Communications. In my previous report, I stated that the observation of the three-fold-symmetric APHE in a trigonal system is interesting and worth publication in a more specialized journal than Nature Physics. I was expecting the authors to soften the main claim and focus on the observation of the three-fold-symmetric APHE itself, but the manuscript resubmitted to Nature Communications still carries the main claim that the APHE is a characteristic transport signature of topological nodal lines (TNLs). This main claim comes with a higher burden of proof.

As I wrote in my previous report, if the APHE shows the peculiar magnetic-field dependence $\sigma_{xy} = \sigma_0 \text{sign}(B) + \alpha B$ mentioned by the authors in their first response, one can reasonably argue that the observed APHE is a characteristic signature of TNLs. In their second response, they now say that the actual situation is complicated and the stated magnetic-field dependence is not really expected to be observed experimentally. In my opinion, this situation makes it difficult to give sufficient support to the main claim, because there is a possibility that the pre-existing Weyl nodes contribute a measurable Hall conductivity as a response to in-plane magnetic fields, depending on the symmetry [see, e.g., H. Wang et al., PRL 132, 056301 (2024)]. Perhaps it would help if the authors provide quantitative comparisons of the expected size of the APHE due to their TNL scenario with all other possible scenarios.

I should note that the authors stated in their second response that, for the occurrence of the APHE, "it is essential to have large contribution of Berry curvature and the only possibility in three-dimensional system are the WN generated by conversion of a topological nodal line". If this statement is legitimately supported theoretically, I think it would make a sufficient support to the main claim. As far as I understand, the manuscript in the present form discusses that TNLs *can* explain the APHE, but the authors did not really state in the manuscript itself that the TNLs are the *only* possible origin of the APHE. If the authors explicitly make this statement in the manuscript with additional argument to support it, I can recommend publication of this manuscript in Nature Communications. As mentioned above, quantitative comparisons with all other possible scenarios may achieve this goal.

Finally, it is important to note that the APHE can be a signature of the TNLs *only when the system has trigonal symmetry*. Even though this condition can be understood from the main text, this important prerequisite is not clearly stated in the abstract. In fact, the current abstract gives the impression that any TNLs should present APHE as its characteristic signature, like the chiral anomaly in Weyl semimetals. This is misleading and should be amended before publication.

Version 1:

Reviewer comments:

Reviewer #1

(Remarks to the Author)

Now I have read the manuscript by Veyrat et al. for the 4th time, along with the response of the authors to my remaining

comments. This time, I find the claims made in the manuscript to be essentially appropriate with reasonable arguments to back them up. In future, when a three-fold symmetric APHE is observed in a trigonal semimetal, one can reasonably argue that the band structure must contain topological nodal lines by citing this paper. This is important for the field of topological materials and I recommend publication of the revised manuscript in Nature Communications.

Before publication, I recommend the authors to consider the following optional comments:

- The reference in line 146 has an error.
- The definition of the angle ϕ shown in Fig. 1d seems to be inconsistent with the definition stated in Methods.
- While the authors focus on the implication of the observation on the topology of the material, I think the three-fold symmetric APHE itself is very unusual and its novelty in the general Hall-effect literature had better be emphasized. For example, there is a recent paper discussing "Parallel-field Hall effect" [Wang et al., PRB 111, L041201 (2025)], where it was shown that such an effect is symmetry-forbidden in most materials and its existence points to a very unusual situation. In my opinion, the present result contributes greatly in this context by showing that the existence of topological nodal lines in trigonal symmetry provides a rare example of such a special situation to give rise to parallel-field Hall effect. Such a discussion would enhance the broad interest of this work.

REVIEWER COMMENTS

Reviewer #1: I have read the revised manuscript and the response of the authors submitted to Nature Communications. In my previous report, I stated that the observation of the three-fold-symmetric APHE in a trigonal system is interesting and worth publication in a more specialized journal than Nature Physics. I was expecting the authors to soften the main claim and focus on the observation of the three-fold-symmetric APHE itself, but the manuscript resubmitted to Nature Communications still carries the main claim that the APHE is a characteristic transport signature of topological nodal lines (TNLs). This main claim comes with a higher burden of proof.

Authors: We appreciate the comment of the Reviewer and in the updated manuscript have toned down the main claims regarding the identification of signatures of TNLs.

Reviewer #1: As I wrote in my previous report, if the APHE shows the peculiar magnetic-field dependence $\sigma_{xy} = \sigma_0 \text{sign}(B) + \alpha B$ mentioned by the authors in their first response, one can reasonably argue that the observed APHE is a characteristic signature of TNLs. In their second response, they now say that the actual situation is complicated and the stated magnetic-field dependence is not really expected to be observed experimentally. In my opinion, this situation makes it difficult to give sufficient support to the main claim, because there is a possibility that the pre-existing Weyl nodes contribute a measurable Hall conductivity as a response to in-plane magnetic fields, depending on the symmetry [see, e.g., H. Wang et al., PRL 132, 056301 (2024)]. Perhaps it would help if the authors provide quantitative comparisons of the expected size of the APHE due to their TNL scenario with all other possible scenarios.

Authors: We can indeed consider a priori the contribution of the pre-existing Weyl node (WN) to two distinct effects leading to similar experimental signatures as the anomalous planar Hall effect (APHE). These effects are i) the intrinsic planar Hall effect, a new orbital mechanism in three-dimensional systems [PRL 132, 056301 (2024)], and ii) APHE at three dimensions, which is under investigation in our work. We demonstrate below that these two contributions are negligible compared to the APHE induced by the topological nodal lines (TNLs) via their conversion into WN. The details of these contributions are as follows:

- i) The intrinsic planar Hall effect. The effect is an orbital anomalous planar Hall effect (APHE) that takes into account the external magnetic field perturbatively, retaining only terms of order $O(E/B)$ with E/B denoting the electric/magnetic field. As also acknowledged by the authors of PRL 132, 056301 (2024), the intrinsic planar Hall effect (IPHE) is forbidden for systems having an out-of-plane rotation axis and should therefore vanish in $t\text{-PtBi}_2$, which exhibits C_{3v} point group symmetry. Conversely, our derivation of the APHE employs a non-perturbative approach where the magnetic field explicitly breaks the material's symmetries, allowing for a trigonal system to exhibit a finite planar Hall response.
- ii) the APHE at three dimensions induced by the preexisting WN. We mentioned in the last version of the manuscript that the APHE generated by the TNL-induced Weyl node has a "much larger amplitude than that which would be obtained solely from the Zeeman-induced displacement of Weyl nodes (see SM)" and that "the Weyl node displacement is proportional to the strength of the applied magnetic field, and remains relatively small at laboratory-accessible fields. The resulting anomalous planar Hall conductance from this displacement is therefore expected to be vanishingly small". Quantitatively, the estimated APHE contribution from the movement of preexisting Weyl nodes (Group G3) in Fig. S14.a of SM S10 amounts approximately to 0.035 S/cm at $B=14\text{T}$, whereas the

contribution of TNL-induced Weyl nodes is found to exceed $4S/cm$ at the same value of the magnetic field, as shown in FIG. 3 of the main text.

We mention now explicitly these two possible sources of APHE in the manuscript:

Quantitatively, the contribution of the twelfold groups to the APHE (see S10) are expected to be more than two orders of magnitude weaker than the contribution of the TNL induced WN. Moreover, we note that another possible origin of the APHE, the orbital intrinsic planar Hall effect, is forbidden in systems with C_{3v} point group symmetry \cite{Wang2024}.

Reviewer #1: I should note that the authors stated in their second response that, for the occurrence of the APHE, “it is essential to have large contribution of Berry curvature and the only possibility in three-dimensional system are the WN generated by conversion of a topological nodal line”. If this statement is legitimately supported theoretically, I think it would make a sufficient support to the main claim. As far as I understand, the manuscript in the present form discusses that TNLs *can* explain the APHE, but the authors did not really state in the manuscript itself that the TNLs are the *only* possible origin of the APHE. If the authors explicitly make this statement in the manuscript with additional argument to support it, I can recommend publication of this manuscript in Nature Communications. As mentioned above, quantitative comparisons with all other possible scenarios may achieve this goal.

Authors: We hope our answer to the previous point and the changes made in the manuscript could convince the reviewer of the pertinence of our analysis.

We have also toned down the main claims regarding the identification of TNL signatures in the revised manuscript in several places in the revised version (see highlighted changes in the revised manuscript).

Reviewer #1: Finally, it is important to note that the APHE can be a signature of the TNLs *only when the system has trigonal symmetry*. Even though this condition can be understood from the main text, this important prerequisite is not clearly stated in the abstract. In fact, the current abstract gives the impression that any TNLs should present APHE as its characteristic signature, like the chiral anomaly in Weyl semimetals. This is misleading and should be amended before publication.

Authors: Indeed, the APHE is restricted to trigonal systems and we have amended the abstract of the manuscript accordingly.

Reviewer #1 (Remarks to the Author):

Now I have read the manuscript by Veyrat et al. for the 4th time, along with the response of the authors to my remaining comments. This time, I find the claims made in the manuscript to be essentially appropriate with reasonable arguments to back them up. In future, when a three-fold symmetric APHE is observed in a trigonal semimetal, one can reasonably argue that the band structure must contain topological nodal lines by citing this paper. This is important for the field of topological materials and I recommend publication of the revised manuscript in Nature Communications.

Before publication, I recommend the authors to consider the following optional comments:

- The reference in line 146 has an error.
- The definition of the angle ϕ shown in Fig. 1d seems to be inconsistent with the definition stated in Methods.
- While the authors focus on the implication of the observation on the topology of the material, I think the three-fold symmetric APHE itself is very unusual and its novelty in the general Hall-effect literature had better be emphasized. For example, there is a recent paper discussing “Parallel-field Hall effect” [Wang et al., PRB 111, L041201 (2025)], where it was shown that such an effect is symmetry-forbidden in most materials and its existence points to a very unusual situation. In my opinion, the present result contributes greatly in this context by showing that the existence of topological nodal lines in trigonal symmetry provides a rare example of such a special situation to give rise to parallel-field Hall effect. Such a discussion would enhance the broad interest of this work.

Authors' response

We would like to thank the reviewer for his comments. We fixed the error in the reference and the fig 1.d and we have now introduced an additional sentence in our manuscript to account for the last comment of the reviewer:

We note that the APHE generally implies finite transversal conductance even for a magnetic field parallel to the electric field, a seldom situation that has already been reported in the case of non-magnetic \cite{Wang2025b} and magnetic materials \cite{Kumar2020}, the latter case exhibiting a threefold symmetry.